# Optical Flow-Based Fast Motion Parameters Estimation for Affine Motion Compensation

Antoine Chauvet [ID] *, Yoshihiro Sugaya [ID] and Tomo Miyazaki [ID] and Shinichiro Omachi [ID]

Graduate School of Engineering, Tohoku University, Sendai, Miyagi 980-8579, Japan;
sugaya@iic.ecei.tohoku.ac.jp (Y.S.); tomo@iic.ecei.tohoku.ac.jp (T.M.); machi@ecei.tohoku.ac.jp (S.O.)
* Correspondence: achauvet@iic.ecei.tohoku.ac.jp

**Abstract:** This study proposes a lightweight solution to estimate affine parameters in affine motion compensation. Most of the current approaches start with an initial approximation based on the standard motion estimation, which only estimates the translation parameters. From there, iterative methods are used to find the best parameters, but they require a significant amount of time. The proposed method aims to speed up the process in two ways, first, skip evaluating affine prediction when it is likely to bring no encoding efficiency benefit, and second, by estimating better initial values for the iteration process. We use the optical flow between the reference picture and the current picture to estimate quickly the best encoding mode and get a better initial estimation. We achieve a reduction in encoding time over the reference of half when compared to the state of the art, with a loss in efficiency below 1%.

**Keywords:** block-based coding; video coding; H.265/HEVC; affine motion compensation

## 1. Introduction

High Efficiency Video Coding (HEVC) [1] is a standard of video coding that is used extensively for High Definition content. It has provided very large gains in coding efficiency compared to previous standards like Advanced Video Coding (AVC) [2].

Most of the efficiency in modern video encoding methods comes from exploiting the similarity between the pictures that form the video sequence, also known as frames. Currently, this works by dividing the current picture into blocks of various sizes and giving them a motion vector and one (or two in case of bi-directional prediction) already decoded pictures to use as source data. The error resulting from prediction, also known as residual, has its entropy further reduced using transforms and quantization. Quantization introduces errors, making the step non-reversible, but it allows for a greatly reduced entropy in the result. Various methods are used to code the prediction modes used and the transformed residual coefficients.

This process works very well when the only changes in the picture can be represented by translations. However, for complex movements, it requires approximating a higher order motion with a translation, leading to prediction error. In most cases, the encoder will decide to use smaller prediction blocks to limit the error for each block, as a larger block would have a more inaccurate motion vector. On the other hand, the high order transform can represent accurately the motion even with a large block. So while additional parameters need to be coded, the reduced amount of blocks means that there are less parameters overall that need to be coded, reducing the cost of coding the motion parameters. Furthermore, this prediction can be more accurate than the translation approximation using many small blocks. The potential of higher order motion models for video coding has been known for a long time, and several papers have demonstrated significant gains, such as Reference [3]. Using affine prediction, they showed an improvement of 6.3% coding efficiency on

sequences using non-translational motion, further increased to 7.6% when using larger blocks up to $128 \times 128$. However, when using smaller blocks, such as $16 \times 16$ like in the previous standard AVC, the gain is reduced to 0.1%. This shows that using large blocks is a critical aspect for higher order motion compensation.

Because affine motion prediction showed very impressive gains on some sequences, it was one of the tools added in the Joint Exploration Model (JEM) [4]. JEM was an experiment to evaluate new proposals for a future encoding standard after HEVC. As affine motion compensation proved it could achieve significant gains, it has been included in the currently being developed future standard Versatile Video Coding (VVC). Several improvements over the original JEM implementation were proposed [5]. While the original implementation supported only a 4-parameter model, it is possible to allow a 6-parameter model as well and let the encoder decide the best model for each block. There are also possible improvements on the entropy coding, based on better motion vector prediction and coding of motion vector differences.

However, this improvement, as most improvements in video encoding, comes with a cost. Most new tools in recent encoding standards work by giving more options to the encoder. For example, allowing larger blocks in HEVC was the source of many improvements in coding efficiency, but this also required much more processing on the encoder side, as to find the best possible block sizes, the encoder needs to try everything. There are 341 possible block partitionings in a given Coding Tree Unit (CTU) [6], and an optimal encoder would need to test all of them to find the most efficient partitioning, which is too demanding for fast encoding, so fast estimation methods are desired. In VVC, the maximum block size is even further increased, increasing even further the amount of possible block partitionings. Higher order motion compensation, like affine motion compensation, is another mode that requires to be evaluated. But the additional encoding time cost is even bigger, as unlike the translation-based motion vectors using two parameters, an affine transform requires six parameters. Classical block-matching approaches do not scale well with more parameters, making them unpractical for this case.

To solve the problem of fast parameter estimation, different methods were designed. In the recent years, the most common implementation for obtaining the parameters is gradient-based. This gradient process is used in many methods, including the affine motion compensation in JEM [7], and methods based on HEVC [8,9]. Typically, the process starts with an initial estimation. The most simple initial estimator is the best translational motion vector, as the motion estimation for translation is performed before the affine motion compensation. If neighbors are available, it is possible to use their affine parameters for the initial estimation. To find a better value, a gradient is computed at the current estimation. The process is repeated either until no improvement is found or a maximum iteration count has been reached. The process is costly as it requires to solve linear equations at each step, but is still much faster than block-matching.

Another method is to reduce the number of parameters of the transform to make the traditional approaches to parameter estimation work in a reasonable time. In Reference [10], the authors have replaced the 4-parameter (also known as zoom&rotation model) transform by two 3-parameter models that can be used depending on the movement. As in most cases, the video is mostly either a rotation or a zoom, it is very common than one of the two parameters is very small or even zero. In those cases, using a model with fewer parameters allows a similar efficiency, and even more in some cases as coding becomes easier. The main drawback is that it requires evaluating the parameters twice. However, this method allows the implementation to use standard block matching techniques that can reuse existing hardware or already implemented methods in software, while solving linear equations in the gradient-based approach requires a completely new implementation. They show a similar time and efficiency compared to Reference [9], but with fewer changes to the existing encoder.

In our proposed method, we decide to use the estimated displacement for each sample in the picture from optical flow to get a faster encoding than the current methods. The displacement can be used to estimate transform parameters for a given affine model. We use this estimation and the variance

of the displacements in a given block to decide what transform model is the most appropriate between zooming, rotation and skipping affine mode parameter estimation entirely. This saves encoding time as fewer affine parameter estimations will be performed. In standard encoding, the parameter estimation for the translation is fast, so optical flow would introduce too much overhead, but the complexity of the affine transform makes motion estimation much slower. We believe the overhead is smaller than the time savings it allows.

In the following section, the current state of the art for affine motion compensation and optical flow will be presented. The methods section will present and explain how our proposed method works. In the results section, we will evaluate the accuracy of the heuristics of the proposed method and compare it with the state of the art.

## 2. Related Works

As mentioned in the introduction, HEVC, by allowing a larger block size compared to AVC, has made affine motion compensation more usable, allowing for very large efficiency gains in sequences that present non-translational motion.

We focus on the implementations on top of HEVC as the proposed implementations in JEM in VVC are not true affine motion compensation, as they compute a standard translational motion vector for $4 \times 4$ subblocks.

### 2.1. Higher Order Motion Prediction Models

In all existing video encoding standards, translation-based prediction is supported. It can be defined mathematically by the following equation:

$$\begin{bmatrix} x' \\ y' \end{bmatrix} = \begin{bmatrix} x \\ y \end{bmatrix} + \begin{bmatrix} v_x \\ v_y \end{bmatrix} \tag{1}$$

where $\begin{bmatrix} x' & y' \end{bmatrix}^t$ represent the coordinates of the points on the reference picture, $\begin{bmatrix} x & y \end{bmatrix}^t$ the coordinates of the points on the current picture, and $\begin{bmatrix} v_x & v_y \end{bmatrix}^t$ the motion vector.

Higher order motion prediction models are models that use more than two parameters to represent motion. While it is possible to define motion models with an arbitrarily high amount of parameters, in practice two models have been used the most: the affine motion model, that uses six parameters, defined by Equation (2), and the zoom and rotation model, that uses four parameters, defined by Equation (3).

$$\begin{bmatrix} x' \\ y' \end{bmatrix} = \begin{bmatrix} a & b \\ c & d \end{bmatrix} \begin{bmatrix} x \\ y \end{bmatrix} + \begin{bmatrix} v_x \\ v_y \end{bmatrix} \tag{2}$$

$$\begin{bmatrix} x' \\ y' \end{bmatrix} = \begin{bmatrix} a & b \\ -b & a \end{bmatrix} \begin{bmatrix} x \\ y \end{bmatrix} + \begin{bmatrix} v_x \\ v_y \end{bmatrix} \tag{3}$$

In these equations, $a$ and $b$ are the affine motion parameters, $v_x$ and $v_y$ are the translational motion parameters. By comparing with Equation (1), we can see that they are very similar, with an additional two or four parameters added.

Tsutake et al. [10] proposed using two 3-parameter models for affine motion compensation to replace the zoom and rotation model, that are defined as follows:

$$\begin{bmatrix} x' \\ y' \end{bmatrix} = \begin{bmatrix} 1+s & 0 \\ 0 & 1+s \end{bmatrix} \begin{bmatrix} x \\ y \end{bmatrix} + \begin{bmatrix} v_x \\ v_y \end{bmatrix} \tag{4}$$

$$\begin{bmatrix} x' \\ y' \end{bmatrix} = \begin{bmatrix} 1 & -r \\ r & 1 \end{bmatrix} \begin{bmatrix} x \\ y \end{bmatrix} + \begin{bmatrix} v_x \\ v_y \end{bmatrix} \tag{5}$$

The two 3-parameter models are simplifications of the 4-parameter zoom and rotation model. The first model, described in Equation (4), sets $b$ to 0 and $a$ to $1+s$, as a value of 0 for $s$ represents a translation. The second model, described in Equation (5), sets $b$ to $r$ and $a$ to 1, so a value of 0 for $r$ represents a translation.

Because it is common that the movement is either zooming or rotation rather than a combination of both, it is common that one of the two affine parameters is much smaller than the other. In this case, reducing the number of parameters will reduce the coding cost of the prediction without losing much accuracy.

Using this dual model option allows for good efficiency, but it requires doing the parameter estimation process twice.

### 2.2. Transform Computation

As seen in the previous equations that represent higher-order motions, they result in a motion vector that depends on the position within the block. While implementations are very good at computing predictions with a constant motion vector (and especially for integer motion vectors as they are simple copy and paste), they are not designed for a constantly changing motion vector.

In the proposed affine motion compensation in JEM [7], this problem is avoided by using constant motion vectors for blocks of $4 \times 4$ samples. However, this also means it is not true affine motion compensation.

In Reference [8,9], the authors suggest doing a $1/16^{th}$ sample interpolation using a eight-tap filter. While it is quite slow, as the gradient method converges quickly towards the optimal value, it does not add too much additional burden to the encoder.

In Reference [10], because the method requires to evaluate more transforms, the interpolation is faster, using the quarter sample interpolation from HEVC and using bilinear interpolation between the four surrounding samples. To avoid the need for computing the interpolation many times, the interpolated samples are stored in a buffer for each reference picture.

### 2.3. Gradient-Based Parameter Estimation

In Reference [5,7–9], a gradient method is used to estimate the affine motion parameters. This method is based on the Newton–Raphson method, which is a method that allows finding the root of a function with an iterative process. The general form is given by the following equation:

$$x_1 = x_0 - \frac{f(x_0)}{f'(x_0)} \tag{6}$$

It is possible to generalize this equation to multi-dimensional problems. With affine motion compensation, we have the following error function:

$$E = \sum_{(x,y)} (org(x,y) - ref(x',y')) \tag{7}$$

where $org(x,y)$ refers to the original value of the sample at coordinates $\begin{bmatrix} x & y \end{bmatrix}^t$ in the current picture, $ref(x,y)$ refers to the sample value at coordinates $\begin{bmatrix} x & y \end{bmatrix}^t$ in the reference picture.

### 2.4. Block-Matching-Based Estimation

Reference [10] use a different method than the others to find the affine parameters. Because they use less parameters, the complexity increase is lower. However, even with only three parameters, the search around neighbors, if using a standard diamond or square pattern, goes from 8 transform computations to 26, and affine prediction is also more costly to compute.

Their idea is to decouple the search for the parameters. As with other methods, they start with an initial estimation based on the classical translation-based motion estimation. Then, they try values in the entire search range, with a step size of $4\Delta$, where $\Delta$ represents the quantization step for the affine parameter. They use the best value they found during this search for the next iterations. The first iteration checks the neighbors at a distance of $2\Delta$, then the second with a distance of $\Delta$. This will give the best affine parameter for the given translation parameters. But the best translation parameters might be different in case of affine prediction, so the second step, the parameter refinement, is performed.

The parameter refinement works by alternating translation parameter refinement and affine parameter refinement. In both cases, the encoder will look for the closest neighbors, at a quarter sample distance for the translation and $\Delta$ for the affine parameter. The refinement stops when either a maximum number of iterations or no more improvement happens.

### 2.5. Motion Parameter Prediction and Entropy Coding

To achieve optimal efficiency when using affine motion prediction, it is important to signal the affine motion parameters with as few bits as possible. Every method uses the same coding as HEVC for the translational parameters, making full use of the motion vector prediction coding.

Reference [9] improves the translational motion vector coding by estimating the change in the translation parameter between blocks. Block-to-block translational shift compensation (BBTSC) corrects the translational shift, allowing merge mode to be used much more often as there is no need to signal the motion vector difference. This results in an improvement of 6% in the tested sequences.

Coding the affine motion parameters is difficult, as it is more difficult to predict them from neighboring blocks. The first limitation is not all blocks are going to use affine prediction, so it may be often necessary to code them without a prediction, but even in the case where a neighbor uses affine prediction, it may use a different reference picture, and scaling the motion parameters is challenging, as simply multiplying every value by the distance ratio does not work. Reference [11] tackles this problem by allowing motion scaling to work on affine parameters. They propose decomposing the transform into separate transforms, for example a rotation and a zoom operation, and scale each matrix appropriately, then combine them again to get the new parameters.

For the quantization, the most common, used in References [3,8,9], is a quantization step of $1/512$. Reference [10] evaluates different quantization step sizes, from $1/16$ to $1/512$. They find that using such a fine quantization step gives no coding efficiency benefit, and that $1/256$ is enough to get the best efficiency. As their method is a semi-exhaustive search, reducing the number of possible values is also good for encoding speed. They also choose to limit the maximum quantized parameter to 16, as higher values are too rare and seldom used.

### 2.6. Optical Flow

Estimating the movement between two pictures has been a subject of research for a long time, as it has numerous applications. For video coding, it is necessary for finding motion vectors, and is often done through computationally expensive methods that check the error for each possible motion vector, with more recent methods improving the search algorithms to keep the encoding time reasonable. In those cases, only the cost for the whole block is considered, so the movement estimation is often not accurate at a more granular level.

However, in many applications, the movement for each pixel is desired. This is typically referred to as optical flow. One of the most famous and popular methods for estimating optical flow is the Lucas-Kanade method [12]. It has been used a lot and gives satisfying results for simple movements. It is quite fast, which is one of the reasons for its popularity. Because it is included in the OpenCV library, it is also very easy to use, while many methods do not release their code, which adds the additional burden of implementation to potential users.

A recent application that also shows potential for video coding is frame interpolation, where by computing the movement for each pixel between the two frames, it is possible to estimate the missing frame with remarkable accuracy, which was demonstrated in EpicFlow [13]. To speed up the process, it is possible to use the motion vectors that are used for encoding the frames as estimators of the motion for a given block, then refine the optical flow to a pixel level, as was proposed in HEVC-Epic [14], which offers a good increase in speed compared to EpicFlow, but is still very slow, taking several seconds for estimating a single frame.

While the computed interpolated frame could be used in encoding with a new kind of prediction, it would make decoding too slow. Decoding needs to be possible on inexpensive hardware to see any large scale adoption.

While the state of the art optical flow methods achieve impressive accuracy, this comes at the cost of increased computation, and depending on the methods the time required varies depending on the picture. When considering hardware implementations and real time constraints, as is the case in encoding, it is important to ensure that the computations will always be bounded as to avoid the need for additional circuitry that will be used only in few cases. In this paper, the optical flow method from Ce Liu [15] was considered because the computation cost varies solely on the size of the input picture and the parameters for the number of iterations.

It also offers very nice properties for hardware implementation, as all the operations are highly parallel in nature, which makes them very easy to implement in hardware. While the software implementation is not parallelized, it would be possible to improve the speed relatively easily.

## 3. Proposed Method

### 3.1. Optical Flow Estimation

For each picture using inter-picture prediction, optical flow is computed using the current picture and the first picture in the reference picture list. While computing it for every picture in the reference picture list leads to better approximations, the required time is much higher, and the proposed method aims to provide good encoding efficiency with a faster encoding than similar methods. For the reference picture, the picture before encoding is used. This offers two advantages: first, this allows optical flow to be computed before the picture is encoded, and second, the motion estimation is more accurate and follows the real movement better, especially when the quantization parameter is large and the reconstructed picture is of lower quality.

After obtaining an approximate displacement for each pixel in the current picture, the estimation is performed for each CTU. As in Reference [3], using smaller blocks improves only slightly the encoding efficiency, but it would require a lot more time. The estimation is based on resolving the linear equation for the 4-parameter model transform with two points in the block.

As the translation parameter can be more accurately estimated with the standard motion estimation technique, only the parameters $a$ and $b$ are considered. Using $x$ and $y$ as the distance between the input points and $x'$ and $y'$ as the distance between the output points, we can estimate $a$ and $b$ with the following equation:

$$a = 1 + s = \frac{xx' + yy'}{x^2 + y^2}$$
$$b = -r = \frac{x'y - xy'}{x^2 + y^2}$$

(8)

To get good results, the points should be far enough apart, so points around the edge of the current block are used. If the points are too close together, cancellation is likely to occur, as the subpixel motion estimation through optical flow is imprecise. To remove the risk of bad estimations from outliers, the values of *a* and *b* are estimated for multiple couples of points, and the median value is retained. When the block is on the edges of the picture and contains pixels outside the reconstructed picture, we cannot compute optical flow on these samples. This happens when the input size is not a multiple of the largest coding block size. In this case, we use samples that are within the reconstructed picture for the computations.

## 3.2. Fast Mode Selection

In other methods, affine prediction is evaluated for each block, which takes a significant amount of time. In Reference [10], there are two affine prediction modes, which take even more time. We propose heuristics to avoid computing all possible modes and save on encoding time.

We decide if affine models should be used over translation first by looking at the variance of the optical flow in a given block. The variance is computed as in Equation (13).

$$\bar{x} = \sum_{i=0}^{N} \sum_{j=0}^{N} \frac{flow_x(i,j)}{N^2} \tag{9}$$

$$\sigma_x^2 = \sum_{i=0}^{N} \sum_{j=0}^{N} \frac{(flow_x(i,j) - \bar{x})^2}{N^2} \tag{10}$$

$$\bar{y} = \sum_{i=0}^{N} \sum_{j=0}^{N} \frac{flow_y(i,j)}{N^2} \tag{11}$$

$$\sigma_y^2 = \sum_{i=0}^{N} \sum_{j=0}^{N} \frac{(flow_y(i,j) - \bar{y})^2}{N^2} \tag{12}$$

$$\sigma_{xy} = \sqrt{\sigma_x^2 + \sigma_y^2} \tag{13}$$

In these equations, $flow_x(i,j)$ and $flow_y(i,j)$ represent the optical flow at the position $(i,j)$. When the resulting variance $\sigma_{xy}$ is very small, translation for the whole block is likely to be very accurate, as every pixel has the same displacement. The opposite case, where the variance is very high, mostly represents large discontinuities in the motion vector we should use to predict the current block. It is very likely that splitting the block into smaller subblocks is preferable.

We decide on two threshold values for these cases, resulting in the following:

1.　translation if $\sigma_{xy} < 0.01$
2.　affine if $0.01 < \sigma_{xy} < 4$
3.　split block if $\sigma_{xy} > 4$

To determine the best threshold values, we ran tests on a few sequences. For the lower bound, 0.01 was determined experimentally to avoid skipping the numerous cases where the best parameter is 1 and the variance would be around 0.05. For the higher bound, we checked the variance of the sequences and values over 1 correlated heavily with object boundaries, but setting the threshold to 1 made the skipping too eager, so we increased it to 4 to allow for some margin of error.

Then, to see which 3-parameter model would fit best, the absolute values of *s* and *r* are compared, and the model corresponding with the highest value is selected. In case neither is bigger than a small threshold, set to a tenth of the minimal non-zero value for the affine parameter, affine motion estimation is skipped for the current block. While in most cases the variance heuristics catch those blocks, some outliers can affect the variance greatly.

To predict values for other pictures in the picture reference list, the displacement is scaled proportionally to the temporal distance between the frames. This approximation is typically accurate enough when the movement stays similar. For example, if the first reference picture is at a distance of 1 and the second at a distance of 2, the displacement values are doubled.

### 3.3. Parameter Refinement

We also propose a very fast refinement algorithm inspired by Tsutake et al. [10]. It is very simplified to reduce the number of iterations. Instead of going over every 4 possible values for the affine parameter, the proposed method encoder only checks the neighbors with a step size of $2\Delta$, refines to $\Delta$ and then refines the quarter pixel translation parameter only once. In case the best value for the affine parameter is zero after the initial neighbor check, the refinement is aborted. In this case, only four affine prediction estimations had to be performed, much fewer than in Tsutake et al. even in the cases of an early abort.

### 3.4. Parallel Processing

The optical flow method requires no encoding information and can be performed while other frames are being encoded. In a typical situation, while the first frame, which has to be Intra, is being encoded, there is enough time for the optical flow computation for the second frame, so if enough CPU cores are available, it can be computed before the need for it arises. If a single frame delay is acceptable, this method will allow saving a significant amount of time in the main encoding loop, which has to iterate over all blocks in order. Even in the case where this one frame delay would be unacceptable, the optical flow method used can be parallelized very well, and as it performs only basic mathematical operations, can easily run on a GPU or dedicated hardware.

## 4. Experimental Results

### 4.1. Testing Conditions

The HEVC reference encoder HM14 [16] is used as the anchor to estimate the Bjøntegaard Delta Bitrate (BD-R) [17] estimated bitrate savings and relative encoding time to compare the various methods.

We used the code from Tsutake et al. [10] to compare our proposed method with the existing state of the art. We also used their implementation of the gradient method from Reference [9] and a 3-parameter variant of the gradient approach that uses the same entropy coding as their method.

We used the same code for the entropy coding and transform calculations. We wrote the parameter estimation of the proposed method to replace theirs. This allows us to compare the parameter estimation process without other variables making the comparison difficult.

We compare our method to Reference [10], their implementation of Reference [9], and the 3-parameter variant of the gradient method.

For the encoding settings, the same settings as Reference [10] are used: The encoding mode is set to Low Delay P, and the quantization parameter (QP) values are $22, 27, 32, 37$.

A total of seven sequences that show various motions were encoded with HM14 [16], Tsutake [10], Heithausen [9], the 3-parameter gradient and the proposed method. The sequences used are from two datasets, the ITE/ARIB Hi-Vision Test Sequence 2nd Edition [18] and Derf's collection [19]. Table 1 lists the sequences that were used, with the sequence number for the sequences from Reference [18]. To compute the encoding time, we used the following formula:

$$\Delta T = \frac{T_{target} - T_{HM_{14}}}{T_{HM_{14}}} \qquad (14)$$

**Table 1.** Video Sequences.

| Sequence Name | Motion |
|---|---|
| Twilight Scene (s215) | Zoom |
| Rotating Disk (s251) | Rotation |
| Fountain (s265) | Rotation |
| Station | Zoom |
| Blue Sky | Rotation |
| Fungus Zoom | Zoom |
| Tractor | Rotation + Zoom |

In the following tables, the encoding time shown is the average over all QP values.

### 4.2. Mode Prediction Accuracy Evaluation

To evaluate the accuracy of our mode selection method, we compared the decisions made with Tsutake [10] with the decision made by the proposed method. We computed how often each affine transform model was used and how accurately the proposed method estimated the correct model. We consider the correct model the one that was used in the final coding, so if a given model was found better than the translation of the full block during the motion estimation phase but was inferior to a split block with different translation, skipping affine is classified as correct choice. We also evaluated the accuracy of the early skipping based on the variance that skips evaluating affine prediction entirely. The results are shown in Table 2.

**Table 2.** Evaluation of mode prediction accuracy of the proposed method compared to Reference [10].

| Sequence Name | Sensitivity | | Best Block Rate | | Correct | Affine Skip | |
|---|---|---|---|---|---|---|---|
| | Rotation[%] | Zoom[%] | Rotation[%] | Zoom[%] | Model[%] | Bad[%] | Missed[%] |
| Station | 14.6 | 90.8 | 0.5 | 33.4 | 89.7 | 3.3 | 70.2 |
| Fountain | 71.2 | 8.0 | 2.0 | 0.4 | 59.8 | 35.3 | 59.8 |
| Fungus Zoom | 0.0 | 99.2 | 0.3 | 35.6 | 98.4 | 0.0 | 100.0 |
| Rotating Disk | 86.9 | 30.9 | 21.6 | 1.3 | 83.9 | 4.5 | 89.7 |
| Blue Sky | 92.8 | 20.4 | 10.2 | 0.7 | 88.2 | 4.5 | 93.6 |
| Tractor | 27.4 | 58.6 | 0.9 | 16.7 | 57.0 | 29.3 | 48.6 |
| Twilight Scene | 30.8 | 38.6 | 0.8 | 1.2 | 35.5 | 34.4 | 83.3 |

Sensitivity represents how often the proposed method predicted this model correctly compared to how often this model was the best when evaluating both. It is calculated with the following formula:

$$\text{sensitivity}_{model} = \frac{\text{TP}_{model}}{\text{TP}_{model} + \text{FN}_{model}} \tag{15}$$

where $\text{TP}_{model}$ is the true positives for a given model (prediction said to use the model and the model was used), and $\text{FN}_{model}$ the false negatives (prediction said to use the other model or to skip while this model was correct).

The best block rate is the percentage of encoded blocks that use that model. The correct model represents how often the proposed method chose the right affine prediction model. It is the weighed average between the sensitivity values for both models, weighed by the prevalence of each model. The skip statistics represent how often the proposed method decided to skip evaluating affine parameters wrongly, and the rate of missed opportunities for skipping affine prediction.

While the accuracy appears to be low for many cases, the accuracy is not weighed with the loss of coding efficiency. While one could measure the efficiency gains estimated for a single block, it is not perfect, as the state of the entropy coder influences the coding of the following blocks. However,

according to our results detailed in the following subsection, the blocks that were predicted incorrectly offered little benefit.

On some sequences where a type of motion is very dominant, like Fungus Zoom where zooming is used much more than rotation, the encoder will often predict the most common model even when it is not the best. This leads to a very low sensitivity for this model, but has a limited effect on the encoding efficiency since that model is not used much. This can be seen with the correct model value that is very high in this case. In the opposite case where rotation is dominant, like Blue Sky, the sensitivity for the zoom models is limited, but it also has a limited effect overall because of the rarity of the other mode.

For some sequences, especially Twilight Scene, the accuracy when compared to the alternative trying every possible transform is very low. However, in this case skipping wrongly according to the reference leads to better results that will be explained in the following subsection. To better investigate why the mispredictions were so common for some sequences, we recorded the affine parameters that were used both when our proposed method predicted accurately and when it gave a bad prediction. The results for the Rotating Disk sequence are presented in Figure 1. It appears that while for all cases smaller (in absolute value) parameters are more common, the wrong predictions have an even higher percentage of small values than the accurate prediction. This suggests that in these cases, the transform brings a smaller gain. As differentiating between the different motions for smaller movements is more complex, the limited accuracy for mode prediction can be understood.

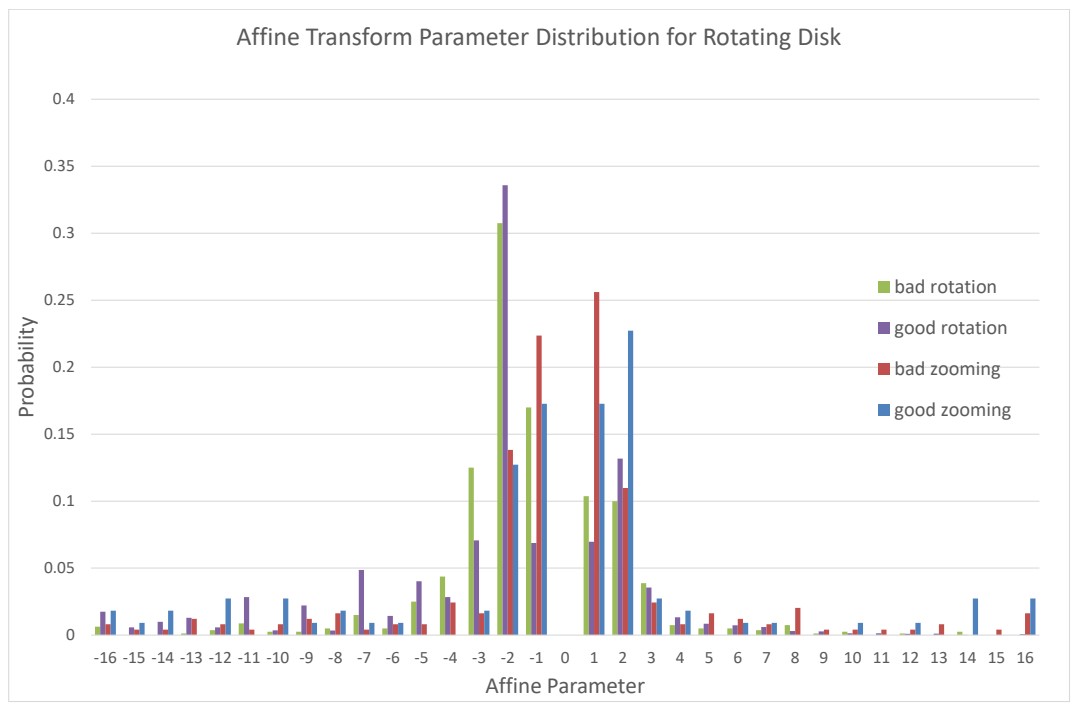

**Figure 1.** Distribution of the affine parameter (quantization levels) in the Rotating Disk sequence for accurately and incorrectly predicted transform models.

### 4.3. Comparison of Variants of the Proposed Method

We evaluated the encoding time and coding efficiency effects of our proposed model prediction, affine mode skip and fast parameter estimation. We compare three variants of the proposed method. The differences between the variants are presented in Table 3.

**Table 3.** Overview of proposed method variants.

| Method Name | Affine Model Prediction | Affine Skip | Fast Parameter Refinement |
|---|---|---|---|
| Model prediction | ✓ | × | × |
| Model + Skip | ✓ | ✓ | × |
| Fast estimation | ✓ | ✓ | ✓ |

Table 4 shows the results of the three variants. While the fast estimation variant is able to achieve the fastest encoding, this comes at the cost of greatly reduced efficiency. If more time is available, on of the other two methods is preferable, as they are able to achieve a better efficiency for a little more time. On some more complex sequences like Tractor, while the encoding speed increases significantly with the fast estimation method, the efficiency is more affected. This sequence is very challenging and the optical flow scaling fails to work on the wheels because of the fast rotation. While optical flow can estimate with some accuracy the motion for the first reference picture, the scaling does not work. As the wheel follows a rotational symmetry, in most cases the correct motion vector does not represent the real movement of the wheel. It will match a similar part of the wheel that has moved less compared to the current picture. Figure 2 illustrates this.

**Table 4.** Comparison of Coding Efficiency and Encoding Speed of the Proposed Method Variants.

| Sequence Name | Model Prediction | | Model + Skip | | Fast Estimation | |
|---|---|---|---|---|---|---|
| | BD-R[%] | $\Delta T$[%] | BD-R[%] | $\Delta T$[%] | BD-R[%] | $\Delta T$[%] |
| Station | −23.75 | 14.54 | −23.68 | 13.25 | −18.03 | 14.32 |
| Fountain | −0.15 | 7.38 | −0.15 | 5.10 | −0.10 | 5.28 |
| Fungus Zoom | −16.18 | 19.77 | −16.42 | 19.92 | −11.38 | 16.77 |
| Rotating Disk | −24.07 | 10.23 | −23.74 | 10.01 | −13.22 | 6.98 |
| Blue Sky | −4.74 | 8.78 | −4.69 | 8.93 | −3.76 | 5.90 |
| Tractor | −3.71 | 8.44 | −3.38 | 5.79 | −1.66 | 5.79 |
| Twilight Scene | −0.26 | 11.05 | −0.42 | 10.65 | −0.27 | 8.01 |
| Average | −10.41 | 11.46 | −10.35 | 10.52 | −6.92 | 9.01 |

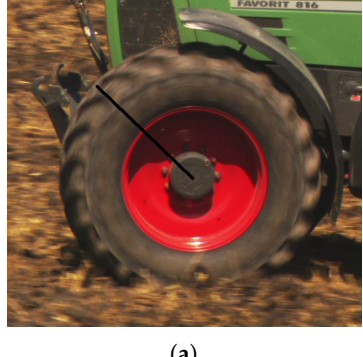 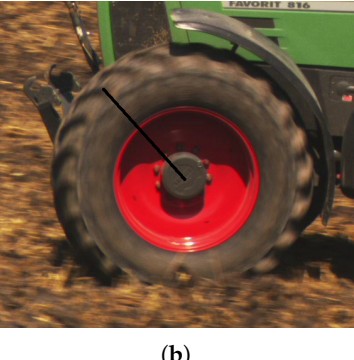 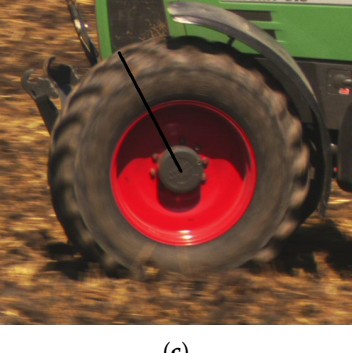

(**a**)　　　　　　　　(**b**)　　　　　　　　(**c**)

**Figure 2.** Three frames extracted from the Tractor sequence: (**a**) Current picture (POC N), wheel angle 136; (**b**) reference picture 0 (POC N-1), wheel angle 132; (**c**) reference picture 1 (POC N-4), wheel angle 118. While the wheel is in a more distant position in (**c**) than (**b**), an acceptable prediction can be done with (**c**) using the wheel rotational symmetry

There are two interesting results for the method using variance to skip affine mode, Twilight Scene and Fungus Zoom, where the efficiency increases with checking the affine mode less often. This happens because the Rate Distortion Optimization (RDO) process is not perfect. While we do not

have a certain explanation, we have two hypotheses: First, not using affine coding at a place where it offered a negligible benefit changed the state of the context in the Context Adaptive Binary Arithmetic Coding (CABAC) enough to improve the coding of future blocks. Second, the tradeoff between quality and bitrate using the Lagrange multiplier is fallible, resulting in a better encoding with an apparently wrong decision. In the present case, for Fungus Zoom, skipping affine resulted in a loss of PSNR of 0.0019dB, for an decrease in bitrate of 0.74% for the quality parameter 27. While the mode accuracy seemed low in Table 2, it seems that it was actually able to remove affine prediction use when it was not beneficial. This suggests that the proposed method is accurate at predicting affine prediction when there is a significant benefit.

On average, the speed is improved by about 1% but in some cases, the required time goes up a little. Some might be caused by the processing of the variance, but we believe it is likely some is from measurement error, as variations of a percent are possible when repeating the same experiment and we only encoded the sequences once for each setting.

Because the efficiency on average decreases only slightly but the encoding time improves, we decided to use this method to compare to the state of the art, as the faster estimation variant reduces efficiency too much.

## 4.4. Comparison with the State of The Art

Table 5 shows how the proposed method compares to Reference [9,10] and the 3-parameter gradient. Table 6 shows the advantages and disadvantages of each method. Each method is able to offer significant improvement for sequences that present affine motion. The current three-parameter model implementations require significant time for encoding, making them difficult to use in practice. Both the proposed and the gradient approach are able to encode sequences with an acceptable overhead. For hardware implementation, gradient methods require many changes, including a more precise sample interpolation scheme, also increasing decoding costs, and a completely different architecture for motion vector parameters estimation. The former is no longer an issue with VVC that made 1/16 sample interpolation the standard for all prediction. However, gradient estimation will still require entirely different circuits. Tsutake [10] is able to provide a solution with minimal hardware changes, but the number of transform evaluations is too important. In our proposed method, there is some additional processing required for the optical flow, but it is possible to implement it at minimal cost, and alternatives for the optical flow method are possible. We believe that overall the total implementation cost is smaller for our proposed method. The last aspect to consider is performance when using content with mostly translation, like the Fountain sequence. In those cases, the proposed method classifies the block as requiring translation, which skips the affine parameter estimation, reducing the encoding speed cost for those sequences where affine prediction offers little encoding efficiency gains. The gradient method performs better than Tsutake here, as it will compute only one transform before giving up, while Tsutake will search many different values first.

On average, the proposed method loses less than 1% in BD-R, but the required encoding time goes down from over 20% in Reference [10] to just over 10%, about half of the time, which is expected from having to evaluate only one of the two affine prediction models, and also skip evaluating both in some cases. In sequences that use mostly translation, like Fountain, skipping many affine prediction blocks reduces the encoding time greatly, from 16.2% overhead to 5.1% with almost no change in efficiency.

However, when comparing with the gradient approach, the encoding time gains are much smaller. Reference [10] shows that their code offers a similar speed to Reference [9], but our experiments show that the gradient approach is much faster. Even the 3-parameter gradient variant that needs to perform the parameter estimation twice is faster than Tsutake. We believe the significant improvement in the speed of the gradient approach comes from the modern compiler used with many optimizations using vector instructions, that were for some reason optimized very well, while the block-matching approach did not get this advantage. However, the block-matching approach still has the advantages described in Reference [10] for hardware implementations as they can reuse more easily existing parts of encoders.

While the optimized version using vector instructions is faster in the software implementation, in hardware it would require a lot more silicon, as there are more operations to perform.

Even with the gradient method being optimized very well for our testing environment, the proposed method is still slightly faster than the 4-parameter gradient method, and significantly faster when compared to the 3-parameter variant. We believe it is possible to use the code of the gradient method to improve our proposed method for both speed and accuracy. On average, the gradient method, even when restricted to fewer parameters, finds slightly better parameters than the block matching approach from Reference [10]. When comparing with the proposed method, the gradient approach offers a better encoding for a limited cost in encoding time but we believe we have a lot of margin left in optimizations.

**Table 5.** Comparison of Coding Efficiency and Encoding Speed.

| Sequence Name | Proposed | | Tsutake [10] | | Gradient [9] | | Gradient 3-Parameter | |
|---|---|---|---|---|---|---|---|---|
| | BD-R[%] | $\Delta T$[%] | BD-R[%] | $\Delta T$[%] | BD-R[%] | $\Delta T$[%] | BD-R[%] | $\Delta T$[%] |
| Station | −23.68 | 13.25 | −24.76 | 24.60 | −25.62 | 12.81 | −24.61 | 22.22 |
| Fountain | −0.15 | 5.10 | −0.16 | 16.20 | −0.24 | 10.49 | −0.17 | 10.07 |
| Fungus Zoom | −16.42 | 19.92 | −16.37 | 26.70 | −15.23 | 17.35 | −16.64 | 23.15 |
| Rotating Disk | −23.74 | 10.01 | −26.77 | 20.42 | −29.28 | 11.24 | −27.28 | 17.06 |
| Blue Sky | −4.69 | 8.93 | −5.13 | 19.18 | −5.52 | 9.23 | −5.78 | 15.52 |
| Tractor | −3.38 | 5.79 | −4.59 | 15.24 | −4.82 | 7.68 | −4.09 | 11.94 |
| Twilight Scene | −0.42 | 10.65 | −0.39 | 20.48 | −0.51 | 10.15 | −0.65 | 16.43 |
| Average | −10.35 | 10.52 | −11.17 | 20.40 | −11.61 | 11.28 | −11.32 | 16.62 |

**Table 6.** Overview of Advantages and Disadvantages of Each Method. ◯ marks when the method is effective, △ when it is acceptable, and × when it is inadequate for this aspect.

| | Proposed | Tsutake [10] | Gradient [9] | Gradient 3-Parameter |
|---|---|---|---|---|
| Encoding Efficiency | ◯ | ◯ | ◯ | ◯ |
| Encoding Speed (affine motion) | ◯ | × | ◯ | △ |
| Encoding Speed (translation) | ◯ | × | △ | △ |
| Hardware implementation | △ | ◯ | × | × |

On some videos, like Blue Sky and Station the efficiency of the proposed method is very close to the existing state of the art, with a increase in time halved compared to Tsutake et al. If there are time constraints, the proposed method can offer superior encoding to HM and close to state of the art while maintaining the encoding time low.

In two sequences, Fungus Zoom and Twilight scene, the efficiency is higher than Tsutake et al., but fails to attain the efficient from the gradient approach using 3 parameters. However, it beats the gradient approach using 4 parameters in the Fungus Zoom case, as the additional unused parameter (rotation being almost inexistent) incurs a coding cost overhead.

Two sequences are very challenging for our proposed method. Tractor was previously mentioned for the limited accuracy for motion estimation, and when compared to the state of the art the effects of the limited accuracy in model estimation are significant. Figure 2 illustrates only one aspect of the challenges in encoding this sequence. Rotating wheel is difficult because of the black background, that optical flow is unable to track, making areas at the edge of the rotating objects hard to estimate. However, as it is a very artificial sequence that is unlikely to appear in more common sequences, we do not believe optimizing for this specific sequence to be sensible.

We can see that while the proposed method does not achieve an efficiency as high as the existing state of the art, it is able to encode in a much faster time, so if time is limited, it could be preferable to use the proposed method as the best compromise between speed and efficiency.

## 5. Conclusions

We presented a solution for the slow encoding when using affine motion compensation by changing the motion estimation algorithm. We proposed three improvements: a fast affine transform model estimation, a skip affine prediction and a fast parameter estimation algorithm. The proposed method is able to predict the correct affine model with good accuracy, and also skip evaluating affine prediction in some cases, saving significant encoding time. When compared to the state of the art, the reduction in bitrate according to the BD-R metric is below 1% on average, with a reduction of the encoding overhead in half compared to Reference [10], and slightly faster than the gradient approach from Reference [9] with less complexity when it comes to hardware implementations. In future work, we plan to investigate ways to make the implementation of the transform faster to reduce further the overhead of affine motion compensation. We also plan to use the optical flow information for block splitting decisions and stop the costly evaluation of smaller blocks when they would bring no benefit.

**Author Contributions:** Methodology, A.C.; Software, A.C.; Validation, Y.S. and T.M.; Writing—Original Draft Preparation, A.C.; Writing—Review & Editing, Y.S. and T.M.; Supervision, S.O.; Project Administration, S.O.; Funding Acquisition, S.O. All authors have read and agreed to the published version of the manuscript.

**Funding:** This study was partially supported by JSPS KAKENHI Grant Number 18K19772 and Yotta Informatics Project by MEXT, Japan

**Acknowledgments:** We would like to thank Tsutake and Yoshida for sharing the code of their method [10], and also their implementation of the other methods they used in their paper.

**Conflicts of Interest:** The authors declare no conflict of interest.

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
