# Peer review of "Optical Flow-Based Fast Motion Parameters Estimation for Affine Motion Compensation"

_applsci, doi:10.3390/app10020729_

Round 1

Reviewer 1 Report

In line 19, it is better to call as a bi-directional prediction than bi-prediction.

In equation 14, what is the impact of two threshold values on the efficacy of the proposed method? How did you arrive at them?

As shown in table 6, proposed method three other methods compared with. It is a positive aspect but still with a limitation for certain types of content.

It is good that you proposed to study further transform faster and block splitting decision making. However, need to be conscious of increasing complexity and time required.

Author Response

Response to Reviewer 1 Comments

Point 1: In line 19, it is better to call as a bi-directional prediction than bi-prediction.

This was changed as requested. Please see highlighted line in the attached pdf on page 1.

Point 2: In equation 14, what is the impact of two threshold values on the efficacy of the proposed method? How did you arrive at them?

We described how we decided the threshold values and the impact when changing them in the following paragraph.

To determine the best threshold values, we ran tests on a few sequences. For the lower bound, 0.01 was determined experimentally to avoid skipping the numerous cases where the best parameter is 1 and the variance would be around 0.05. For the higher bound, we checked the variance of the sequences and values over 1 correlated heavily with object boundaries, but setting the threshold to 1 made the skipping too eager, so we increased it to 4 to allow for some margin of error.

Please see the highlighted text on page 7 in the attached pdf.

Point 3: As shown in table 6, proposed method three other methods compared with. It is a positive aspect but still with a limitation for certain types of content.

We described in more details the advantages and disadvantages of each method. We mention the limitations of the various methods for some content, notably sequences with mostly translation. Please see the modified table with a differentiation of encoding speed depending on the motion type on pages 12 and 13 in the attached pdf.

Each method is able to offer significant improvement for sequences that present affine motion.
The current three-parameter model implementations require significant time for encoding, making them difficult to use in practice.
Both the proposed and the gradient approach are able to encode sequences with an acceptable overhead.
For hardware implementation, gradient methods require many changes,
including a more precise sample interpolation scheme, also increasing decoding costs,
and a completely different architecture for motion vector parameters estimation.
The former is no longer an issue with VVC that made 1/16 sample interpolation the standard for all prediction.
However, gradient estimation will still require entirely different circuits.
Tsutake is able to provide a solution with minimal hardware changes, but the number of transform evaluations is too important.
In our proposed method, there is some additional processing required for the optical flow,
but it is possible to implement it at minimal cost, and alternatives for the optical flow method are possible.
We believe that overall the total implementation cost is smaller for our proposed method.
The last aspect to consider is performance when using content with mostly translation, like the Fountain sequence.
In those cases, the proposed method classifies the block as requiring translation, which skips the affine parameter estimation,
reducing the encoding speed cost for those sequences where affine prediction offers little encoding efficiency gains.
The gradient method performs better than Tsutake here, as it will compute only one transform before giving up,
while Tsutake will search many different values first.

Point 4: It is good that you proposed to study further transform faster and block splitting decision making. However, need to be conscious of increasing complexity and time required.

We mention the complexity issues in the reply to Point 3.

Reviewer 2 Report

The research paper presents a solution for the slow encoding when using affine motion compensation by hanging the motion estimation algorithm. The authors proposed three improvements: a fast affine transform, model estimation, a skip affine prediction, and a fast parameter estimation algorithm.

Note: Eq.(4) does not describe an equation. It could be improved/re-written. 

Author Response

Response to Reviewer 2 Comments

Point 1: Note: Eq.(4) does not describe an equation. It could be improved/re-written.

I assume you meant Equation 14. You are right, the equation form is not appropriate for this, so we changed it to an enumeration with the three choices, with a paragraph detailing the thresholds and the reasons behind them. Please see in the attached pdf.
